# The impact of comprehensive geriatric assessment on postoperative outcomes in elderly surgery: A systematic review and meta-analysis

Lin Chen[1]*, Wei Zong[2], Manyue Luo[3], Huiqin Yu[1]

1 Anesthesia and Surgery Department, Chengdu Second People's Hospital, Chengdu, Sichuan, China,
2 Department of Critical Care Medicine, First Affiliated Hospital of Xuzhou Medical University, Xuzhou, Jangsu, China, 3 Endocrinology and Metabolism Department, Changsha People's Hospital, Changsha, Hunan, China

* CL940483030@hotmail.com

## Abstract

**Data Availability Statement:** All relevant data are within the manuscript and its Supporting Information files.

### Introduction

The elderly population experiences more postoperative complications. A comprehensive geriatric assessment, which is multidimensional and coordinated, could help reduce these unfavorable outcomes. However, its effectiveness is still uncertain.

### Methods

We searched multiple online databases, including Medline, PubMed, Web of Science, Cochrane Library, Embase, CINAL, ProQuest, and Wiley, for relevant literature from their inception to October 2023. We included randomized trials of individuals aged 65 and older undergoing surgery. These trials compared comprehensive geriatric assessment with usual surgical care and reported on postoperative outcomes. Two researchers independently screened the literature, extracted data, and assessed the certainty of evidence from the identified articles. We conducted a meta-analysis using RevMan 5.3 to calculate the Odds Ratio (OR) and Mean Difference (MD) of the pooled data.

### Results

The study included 1325 individuals from seven randomized trials. Comprehensive geriatric assessment reduced the rate of postoperative delirium (28.5% vs. 37.0%; OR: 0.63; CI: 0.47–0.85; I2: 54%; P = 0.003) based on pooled data. However, it did not significantly improve other parameters such as length of stay (MD: -0.36; 95% CI: -0.376, 3.05; I2: 96%; P = 0.84), readmission rate (18.6% vs. 15.4%; OR: 1.26; CI: 0.86–1.84; I2: 0%; P = 0.24), and ADL function (MD: -0.24; 95% CI: -1.27, 0.19; I2: 0%; P = 0.64).

**Funding:** The author(s) received no specific funding for this work.

**Competing interests:** The authors have declared that no competing interests exist.

## Conclusions

Apart from reducing delirium, it is still unclear whether comprehensive geriatric assessment improves other postoperative outcomes. More evidence from higher-quality randomized trials is needed.

## Introduction

The World Health Organization estimates that 1 billion people worldwide were 60 years of age or older in 2019. This figure is projected to increase to 1.4 billion by 2030 [1]. As the population ages, a growing number of elderly individuals will need surgical care [2]. However, most older individuals have many chronic illnesses that hinder their ability to heal and function [3]. Additionally, non-disease-related issues can complicate surgery and the healing process. These issues include frailty, multiple medications, degenerative organ changes, poor nutrition, and cognitive decline [4–6].

Elderly individuals face higher surgical mortality, longer hospital stays, more frequent in-hospital adverse events (such as delirium, pressure ulcers, urinary incontinence, and functional decline), and more readmissions after discharge compared to younger patients [7–10]. Therefore, it is essential for surgical practitioners to identify factors that could lead to unfavorable outcomes in senior patients through timely assessments [11]. However, the current standard evaluations have significant limitations. They cannot measure the body's reserve and capacity for compensation because they mainly focus on specific organ systems or are highly subjective [12].

Comprehensive geriatric assessment (CGA) involves a multidisciplinary team of geriatric physicians, nurses, anesthesiologists, surgeons, physiotherapists, occupational therapists, and nutritionists [13]. This approach uses a multidimensional perspective to evaluate the physical condition, functional status, mental well-being, and social environment of older adults. Based on this evaluation, a comprehensive and coordinated plan is created to improve the quality of life for elderly patients [14,15].

It is logical that older individuals undergoing elective surgery may benefit from a CGA. This study aimed to compare the impact of CGA with conventional treatment on unfavorable outcomes in older patients having elective surgery. A methodical search and meta-analysis of relevant literature were conducted for this investigation.

## Methods

The study was reported in the International Prospective Systematic Reviews Register (PROS-PERO) (registration number CRD42023478608) and conducted in compliance with the PRISMA standards [16] for systematic reviews and meta-analyses.

### Search strategy

The databases searched included Medline, PubMed, Web of Science, Cochrane Library, Embase, CINAHL, ProQuest, and Wiley Online Library. Both published and unpublished publications were included. A combination of free text and Medical Subject Headings (MeSH) was used to find relevant material, with modifications made for specific databases. The search criteria were based on PICOS and included terms such as Elderly, Comprehensive geriatric

evaluation, Postoperative, and Study types. The search period was up to October 2023. Detailed electronic search strategies are provided in the supplemental material.

## Literature inclusion criteria

(i) Patients over 60 years old undergoing elective non-cardiac high-risk surgery served as research participants; (ii) The study employed a randomized controlled trial design. The control group received standard care (standard preoperative evaluation), while the intervention group used CGA as a fundamental component of geriatric care; (iii) The study reported at least one of the following postoperative outcomes: length of stay, 30-day readmission rate, 30-day mortality, and any other postoperative complications (such as delirium, pressure ulcers, urinary incontinence, and functional decline). Exclusion criteria: (i) Studies that used CGA solely as a risk assessment tool for postoperative unfavorable outcomes; (ii) Studies that did not use a comprehensive multidomain evaluation and optimization plan, but only evaluated one CGA field, such as nutritional status; (iii) Non-English publications; (iv) Studies focused on outpatient or emergency surgery or non-elderly populations; (v) Study registration protocols.

## Data extraction

Two researchers independently screened the literature based on the title, abstract, and full text. They retrieved data and evaluated the risk of bias. All gathered information was documented uniformly according to the content of the literature. In cases of disagreement, the two researchers consulted a third researcher or discussed the issue together to decide whether to include the content.

## Quality assessment

The Cochrane Handbook [17] was used to evaluate the quality of the listed randomized controlled trials, focusing on: (i) Randomization sequence generation; (ii) Allocation concealment; (iii) Blinding of outcome assessment; (iv) Selection reporting bias; (v) Completeness of outcome data; (vi) Other sources of bias.

The research group decided to exclude blinding of researchers and participants from the quality assessment, as it was deemed impractical during the CGA intervention. The risk categories were rated as 'insufficient,' 'sufficient,' or 'don't know'. Studies that did not meet the quality standards were excluded.

## Data analysis

RevMan 5.3 was used for the meta-analysis. A weighted odds ratio (OR) was used for count data, and a weighted mean difference (WMD) for measurement data if the same instruments were used. Otherwise, a standardized mean difference (SMD) was used. All analyses were performed with 95% confidence intervals (CI). Heterogeneity between studies was assessed using the $I^2$ statistic and the chi-square test.

A fixed-effects model was used if there was no statistical heterogeneity ($P > 0.1$, $I^2 < 50\%$). A random-effects model was used if there was statistical heterogeneity but no clinical heterogeneity ($P < 0.1$, $I^2 \geq 50\%$). If $I^2$ was inconsistent with P ($P < 0.1$, $I^2 < 50\%$ or $P > 0.1$, $I^2 \geq 50\%$), a model with $I^2$ as the reference was used. If the cause of heterogeneity could not be identified, a descriptive analysis was employed.

Egger's test was used to evaluate publication bias. The impact of each study on heterogeneity was investigated by removing studies one at a time and recalculating heterogeneity.

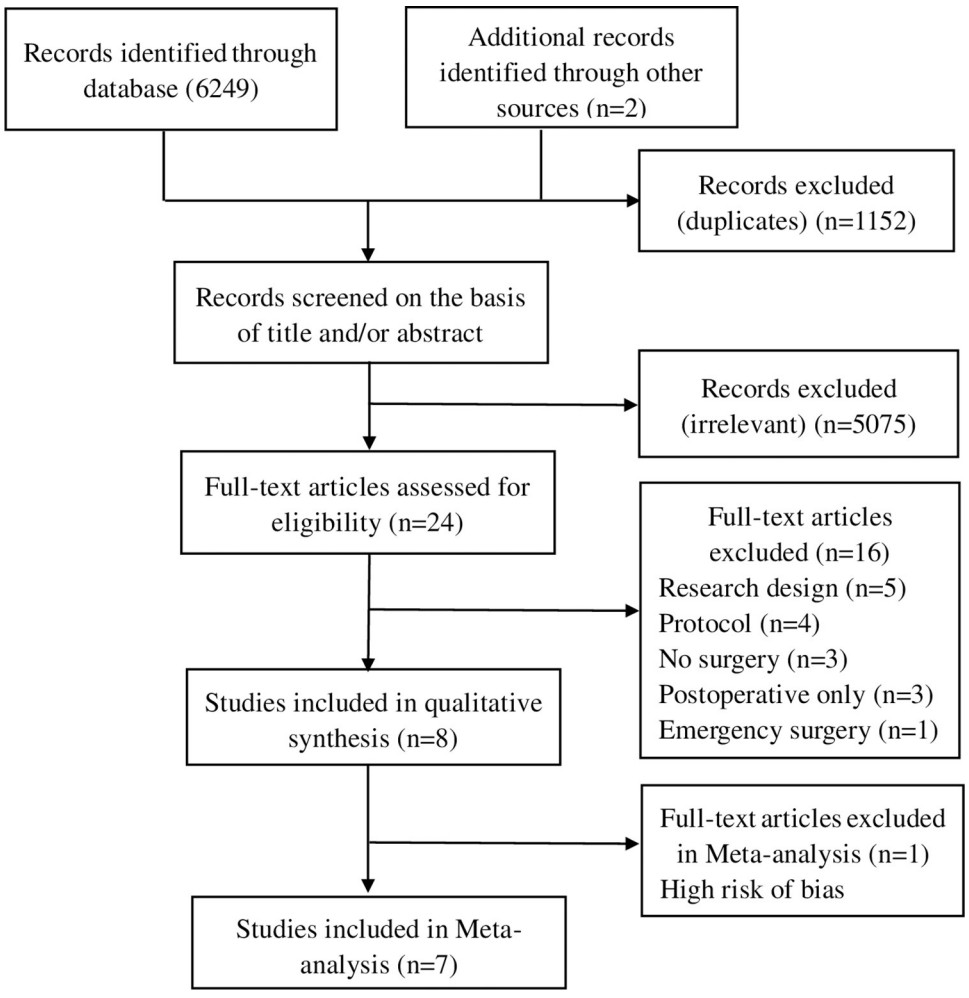

**Fig 1. Flow chart.**

## Results

### Search results

A total of 6249 items were found. After removing 1152 duplicates using EndNote, 24 articles were initially included. Reviews, clinical record comparisons, case reports, and other literature that did not meet the inclusion criteria were removed based on their titles and abstracts. After a closer examination of the full text, one low-quality article and sixteen that did not meet the inclusion criteria were excluded. Ultimately, 7 publications [18–24] were included in the meta-analysis (Fig 1).

Inter-rater reliability was assessed using Cohen's kappa when selecting studies. The findings showed satisfactory agreement between the evaluators (Kappa = 0.76; 95% CI: 0.56, 0.96; p < 0.001). An overview of patient characteristics from the seven trials is compiled in Table 1, and the quality evaluation results are shown in Table 2.

### Study characteristics

Seven studies, published between 2012 and 2022, comprise this qualitative analysis. The studies were conducted in the Netherlands, China, Sweden, Norway, and the United Kingdom. The

**Table 1. The characteristics of qualified researches.**

| Author | Year | Country | Surgery type | Intervention Contents | Sample size | | Age | | measures |
|--------|------|---------|--------------|----------------------|-------------|---|-----|---|----------|
| | | | | | Intervention | Cotrol | Intervention | Cotrol | |
| Watne [18] | 2014 | Norway | Hip fracture | 1.CGA<br>2.Medication reviews, Early and intensive mobilization, Optimizing pre- and postoperative nutrition, Early discharge planning | 163 | 166 | 84 (55 to 99) | 85 (46 to 101) | abcdefgh |
| Partridge [19] | 2017 | England | Vascular | 1.CGA<br>2.Evaluation and optimization according to Cognition, Anaemia, Cardiac, Frailty | 85 | 91 | 75.5±6.6 | 75.5 ±6.3 | bchij |
| Ommundsen [20] | 2018 | Norway | Colorectal cancer | 1.CGA consisted of ADL,Use of Medication, Comorbidity,Nutritional status,Cognition, Depression | 52 | 62 | 78.2±7.4 | 78.8 ±7.8 | chjkl |
| Hempenius [21] | 2013 | Netherlands | Solid tumour | 1.CGA<br>2.Preoperative geriatric consultation and postoperative monitoring,Individual treatment for delirium,Ddaily visits,advice on encountered problems | 127 | 133 | 77.45±6.72 | 77.63 ±7.69 | abdjmno |
| Zhu [22] | 2022 | China | Hip fracture | 1.CGA<br>2.Assessment of basic medical diseases,Nutrition, Physical activity,Cognitive function,Frailty | 70 | 85 | 79.5 | 81.8 | bcfpq |
| Stenvall [23] | 2012 | Sweden | Hip fracture | 1.CGA<br>2.Individualized care planning and rehabilitation, active prevention, detection and treatment of postoperative complications | 28 | 36 | 81.0±5.8 | 83.2 ±6.4 | bcfj |
| Hempenius [24] | 2016 | Netherlands | Solid tumour | 1.CGA<br>2.Preoperative geriatric consultation and postoperative monitoring,Individual treatment for delirium,Ddaily visits,advice on encountered problems | 106 | 121 | 77.37±6.88 | 77.42 ±7.71 | adhfmno |

(a)Cognitive function; (b)Delirium; (c)Length of stay; (d)Mortality; (e)Mobility;(f)ADL function;(g) Weight changes; (h)Re-admissions; (i)New co-morbid diagnoses; (j) Postoperative medical and surgical complications such as pneumonia,wound infection.et,al; (k)Reoperation within 30 days; (l)Survival at 30 days and 3 moths; (m)Care dependency; (n)quality of life;(o)Return to an independent preoperative living situation;(p)Operation rate at 48 h;(q) Preoperative waiting time.

**Table 2. Quality assessment.**

| | A | B | C | D | E | F |
|--|---|---|---|---|---|---|
| Watne | + | + | + | − | + | − |
| Partridge | + | − | + | − | + | − |
| Ommundsen | + | − | + | − | + | − |
| Hempenius | + | DK | DK | − | + | − |
| Zhu | + | DK | DK | − | + | − |
| Stenvall | + | + | + | − | + | − |
| Hempenius | + | + | + | − | − | − |

+, sufficient; −, insufficient;DK, don't know;A,Was a random grouping method employed?;B, Was the allocation between the treatment/intervention group and the control group concealed?;C, Were blind methods applied for outcome evaluators during the intervention process?; D, Were any selective reporting of research results? E, Were the outcomes adequately recorded for the study purpose?;F, Were important confounding and effect-modifying variables taken into account in the design and/or analysis?

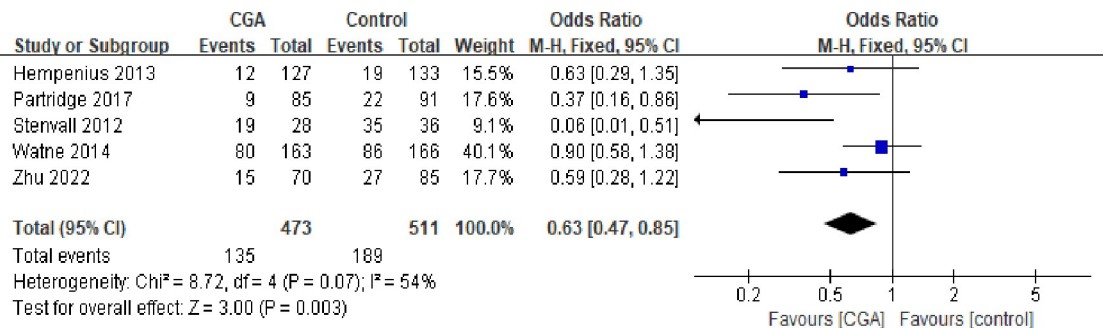

**Fig 2. Forest plot displaying a meta-analysis of the delirium.**

average age of the participants was between 75.5 and 85 years old. The surgeries studied included three hip fractures, two solid tumors, and colorectal and vascular cancer. Sample sizes ranged from 64 to 329 participants.

The outcome variables examined included cognitive function, delirium, length of stay, mortality, mobility, activities of daily living (ADL), weight changes, readmissions, new comorbid diagnoses, reoperation within 30 days, survival at 30 days and 3 months, care dependency, quality of life, return to an independent preoperative living situation, operation rate within 48 hours, preoperative waiting time, and postoperative complications such as pneumonia and wound infections (Table 1).

## Effect sizes

**Delirium.** Postoperative delirium prevalence was studied in five trials [18,19,21–23] involving 473 elderly patients in the intervention group and 511 elderly patients in the control group. The analysis found that the rate of delirium was significantly lower in the CGA group compared to the control group (28.5% vs. 37.0%; OR: 0.63; CI: 0.47–0.85; $I^2$: 54%; P = 0.003) (Fig 2).

**Length of stay.** Six trials examined the impact of CGA on the length of hospital stay following surgery. However, Ommundsen et al.'s study [20] was excluded because it only provided the median and lacked additional data, making meta-analysis impossible. Ultimately, five studies [18,19,21–23] with 473 patients in the intervention group and 511 patients in the control group were included. The findings indicated that although the intervention group's average hospital stay was shorter than the control group's, this difference was not statistically significant (mean difference: -0.36; 95% CI: -0.376, 3.05; $I^2$: 96%; p = 0.84). A sensitivity

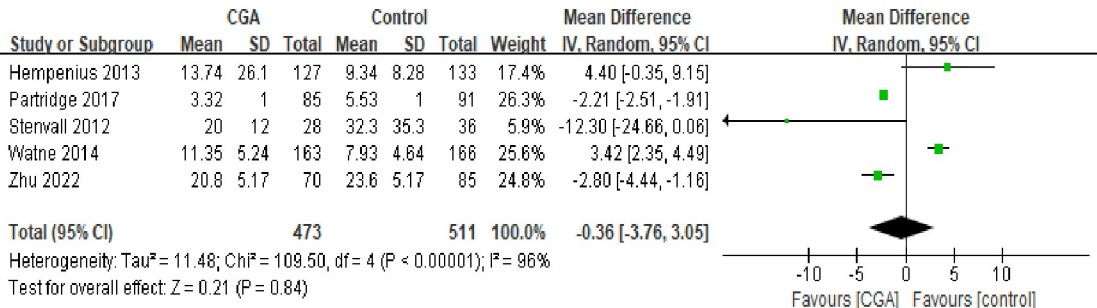

**Fig 3. Forest plot displaying a meta-analysis of the length of stay.**

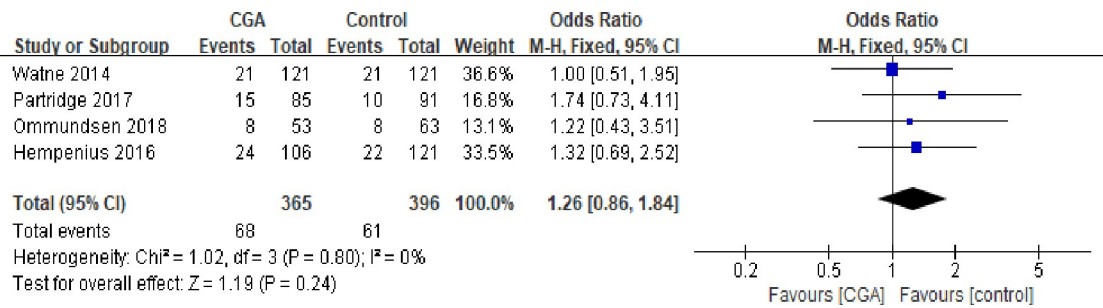

**Fig 4. Forest plot displaying a meta-analysis of the re-admissions.**

analysis revealed that the pooled estimates were not significant due to excessive heterogeneity (Fig 3).

**Re-admissions.** Four studies [18–20,24] reported the rate of readmissions within 30 days following surgery, involving 396 patients in the control group and 365 in the intervention group. The short-term readmission rate was slightly higher in the intervention group (18.6%) compared to the control group (15.4%). However, the pooled data showed no statistically significant difference between the two groups (18.6% vs. 15.4%; OR: 1.26; CI: 0.86–1.84; $I^2$: 0%; P = 0.24) (Fig 4).

**Activities of daily living (ADL) functioning.** Four studies provided data on ADL function, all using the Nursing Dependency Scale [25] for assessment. A meta-analysis was conducted on three trials [18,22,23], including 215 patients in the intervention group and 236 in the control group, since Hempenius et al.'s study [24] only provided baseline ADL scores. The analysis did not reveal any significant differences in ADL between the two groups (mean difference: -0.24; 95% CI: -1.27, 0.19; $I^2$: 0%; p = 0.64) (Fig 5).

**Other postoperative outcomes.** Although not statistically significant, other outcome measures discussed included quality of life [21,24], care reliance [21,24], mortality [18,21,24], mobility [18], reoperation within 30 days [20], survival at 30 days and 3 months [20], and cognitive function [18,21,24]. Significant improvements were observed in the operation rate within 48 hours [22] (p < 0.001), preoperative waiting time [22] (p < 0.001), wound infection [19] (p = 0.032), cardiac complications [19] (p = 0.001), bowel and bladder problems [19] (p = 0.003), and tract infections [23] (p = 0.001). However, the intervention group experienced a significantly greater number of new comorbid diagnoses [19], including cognitive impairment, chronic renal illness stage 3 or above, and chronic obstructive pulmonary disease (p < 0.001). Additionally, two studies [21,24] on postoperative recovery were conducted; one [21] found that a higher proportion of patients in the control group reverted to their preoperative way of life (OR: 1.84; 95% CI: 1.01–3.37).

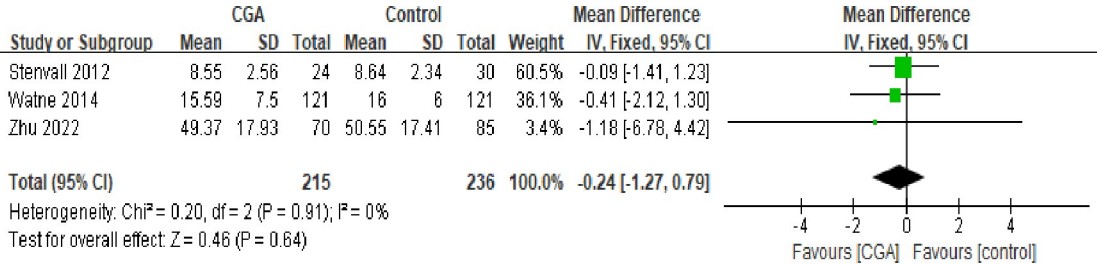

**Fig 5. Forest plot displaying a meta-analysis of the ADL function.**

**Table 3. Risk stratification methods for elderly surgical patients.**

| Method | Description | Focus Area |
|---|---|---|
| Charlson Comorbidity Index (CCI) | Scores patients based on a list of 19 comorbid conditions, predicting 10-year survival in patients undergoing surgery. | Comorbidities |
| Modified Frailty Index (MFI) | Uses 11 of the 70 items from the Canadian Study of Health and Aging Frailty Index to predict postoperative outcomes. | Frailty assessment |
| ASA Physical Status Classification | A scale that assesses and classifies the fitness of patients before surgery. | Overall physical status |
| Surgical Apgar Score (SAS) | Provides a score based on intraoperative parameters to predict postoperative outcomes. | Intraoperative risk |
| Elixhauser Comorbidity Index (ECI) | Identifies comorbidities that might alter the risk of mortality and morbidity in hospital settings. | Comorbidities |
| Comprehensive Geriatric Assessment (CGA) | A multidimensional, interdisciplinary diagnostic process to determine the medical, psychological, and functional capabilities of elderly patients to develop a coordinated and integrated plan for treatment and long-term follow-up. | Overall health assessment |

**Publication bias.** Egger's test was used to ass1ess publication bias in delirium studies, and the results suggest that publication bias may be a cause for concern (z = 2.85, p = 0.004).

## Discussion

1.This article reviewed randomized controlled studies on the impact of Comprehensive Geriatric Assessment (CGA) on the postoperative prognosis of elderly patients undergoing elective surgery. Seven studies met the inclusion criteria. They all optimized perioperative care by managing cognitive impairment, frailty, and comorbidities. This was achieved through preoperative assessments and targeted interventions, such as medication reviews, nutritional support, and consultations. These studies demonstrate the importance of CGA as a preventive measure to improve postoperative outcomes.

Unfortunately, among the many outcome indicators considered, only the incidence of postoperative delirium showed that the intervention was effective, which differed from the expected conclusion. This review conducted a meta-analysis only on postoperative delirium, length of stay, ADL function, and readmission rate because few studies reported additional indicators, limiting the pooled results.

2. A more comprehensive assessment of perioperative risk may have led more patients in the intervention group to choose conservative treatment, which could limit the effectiveness of CGA. For example, two studies [18,21] reported no significant differences in length of stay or care dependence, but found that more intervention patients were transferred to nursing homes for better rehabilitation care. The lack of follow-up observations may result in an inaccurate estimate of the intervention's effect.

The long duration of the CGA and its reliance on geriatric knowledge might contribute to inadequate recognition of patient risk factors. This is especially true for patients who are cognitively impaired, take multiple medications, have multiple chronic conditions, or lack social support. Additionally, the lack of a longitudinal relationship between the evaluator and the patient may prevent the evaluator from addressing undiagnosed cognitive impairments or psychological problems. These issues can reduce the accuracy of assessing the benefits of interventions for outcomes influenced by multiple factors, such as readmission rates and postoperative complications.

3. During a review of the literature, it was found that most studies focused only on the assessment part of CGA and did not include specific care models or management plans for modifiable risk factors aimed at improving postoperative outcomes. Therefore, these studies did not provide a complete CGA.

More than half of the interventions in this review were multidisciplinary care pathways [18,20,22,23]. The other two were the Proactive Care of Older People Having Surgery (POPS) [19] and the Liaison Intervention in Frail Elderly (LIFE) pathways [21,24]. Additional reported models included the Perioperative Optimization of Senior Health (POSH) [26], the Hospital Elder Life Program (HELP) pathway [27], and the Person-Centered Care (PCC) pathway [28]. However, no relevant randomized controlled trials were identified.

4. In surgical care for elderly patients, many preoperative risk stratification methods are similar to CGA. All aim to evaluate surgical risks for senior citizens, allowing medical professionals to tailor preventative measures more effectively (Table 3). Common methods include the Charlson Comorbidity Index (CCI) [29], the Modified Frailty Index (MFI) [30], the ASA Physical Status Classification [31], the Surgical Apgar Score (SAS) [32], and the Elixhauser Comfort Index (ECI) [33].

However, unlike CGA, these methods focus on clinical and physiological parameters that directly affect surgical risk, helping to decide whether patients should proceed with surgery and what level of care they need afterward. They provide faster evaluations but may miss subtle differences that a complete CGA can capture. CGA has a broader scope and is more widely used to assess the overall health status of patients. These aspects may not directly affect surgery but are crucial for patients' overall health and postoperative recovery.

5. Bias is inevitable. First, the potential for bias increased when the same researchers were involved in both the intervention and control groups, as the included studies were single-center and could not blind investigators and participants during the CGA intervention. Second, a time gap between pre- and post-operative evaluations could introduce extraneous variables [34]. Some studies addressed this by shortening the observation period, leading researchers to select quickly examinable items. However, this may lead to publication bias. For example, studies on immediate postoperative delirium may be published earlier than those on long-term cognitive effects [35]. Insufficient follow-up time can also result in false-negative outcomes.

Third, this review is limited to English-language research, which may introduce bias. Despite following guidelines for meta-analyses of epidemiological observational studies, there remains some subjectivity in the consensus protocol for assessing the quality of included studies.

6. Even so, the value of preoperative CGA and collaborative geriatric management in helping medical professionals address the complexities of elderly care is undeniable [36–39]. However, evaluating preoperative CGA is challenging, as current information is inconclusive and requires definitive trials. These trials must comply with the CONSORT guidelines [40], using rigorous, randomized, controlled designs and outcome measures.

The research should fully implement CGA interventions, including optimization and assessment, select a specific strategy model, and standardize the evaluation process and instruments. Additionally, researchers should identify the specific elements needing intervention and the subtle observational variables affected by CGA. It is crucial to find suitable individuals who are likely to benefit from CGA.

## Conclusion

Current evidence suggests that few studies have thoroughly applied CGA to surgical patients. It is unclear how CGA affects postoperative outcomes, other than delirium, in older patients undergoing elective non-cardiac surgery. More research through higher-quality randomized controlled trials is needed.

## Supporting information

**S1 File. PRISMA checklist.**
(DOCX)

**S2 File. Detailed search criteria in PubMed.**
(DOCX)

## Author Contributions

**Conceptualization:** Lin Chen.

**Data curation:** Lin Chen, Wei Zong, Manyue Luo.

**Formal analysis:** Wei Zong, Manyue Luo.

**Project administration:** Lin Chen, Huiqin Yu.

**Software:** Lin Chen, Manyue Luo.

**Supervision:** Huiqin Yu.

**Validation:** Huiqin Yu.

**Writing – original draft:** Lin Chen.

**Writing – review & editing:** Wei Zong.

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
