## [Decision Letter · Decision Letter 0]

10 May 2024

PONE-D-24-12165The impact of comprehensive geriatric assessment on postoperative outcomes in elderly surgery: a systematicreview and meta-analysisPLOS ONE

Dear Dr. Chen,

Thank you for submitting your manuscript to PLOS ONE. After careful consideration, we feel that it has merit but does not fully meet PLOS ONE’s publication criteria as it currently stands. Therefore, we invite you to submit a revised version of the manuscript that addresses the points raised during the review process.

We look forward to receiving your revised manuscript.

Kind regards,

Barry Kweh

Academic Editor

PLOS ONE

Journal Requirements:

[ ]. 

3. We note that you have referenced (unpublished) on page 3, which has currently not yet been accepted for publication. Please remove this from your References and amend this to state in the body of your manuscript: (ie “Bewick et al. [Unpublished]”) as detailed online in our guide for authors

Additional Editor Comments:

The authors have conducted an interesting and robust meta-analysis. There needs to be greater emphasis on inclusion and exclusion criteria as well as a more thorough risk of bias assessment using PRISMA-endorsed instruments. The discussion also requires a greater overview of the literature in both a tabulated and written format. This should demonstrate and discuss other means of risk stratification in elderly patients undergoing surgery such as the CCI or MFI.

Reviewers' comments:

Reviewer's Responses to Questions

**Comments to the Author**

1. Is the manuscript technically sound, and do the data support the conclusions?

Reviewer #1: Partly

Reviewer #2: Yes

2. Has the statistical analysis been performed appropriately and rigorously? 

Reviewer #1: Yes

Reviewer #2: Yes

3. Have the authors made all data underlying the findings in their manuscript fully available?

Reviewer #1: Yes

Reviewer #2: Yes

4. Is the manuscript presented in an intelligible fashion and written in standard English?

Reviewer #1: No

Reviewer #2: Yes

5. Review Comments to the Author

Reviewer #1: The authors conducted a systematic review, which is an interesting topic to choose, but some issues need to be addressed before publication:

1. Don't use acronyms for the first time. It's confusing.

2. Are the methods of CGA exactly the same? In the included original studies, if not, the authors may not be able to conduct a combined analysis.

3. The reasons for publication bias and high heterogeneity should be explained in the discussion.

4. Full text language needs to be modified.

Reviewer #2: The study will definitely contribute to the wealth of medical knowledge and it will impact geriatric patient management. The search was broad, the article selection was scientifically done ,methodology was good, and their distribution of the articles was wide.The data extraction was appropriate, result was well analysed and the conclusion was in tandem with the results.

The references are appropriate.

6. PLOS authors have the option to publish the peer review history of their article (what does this mean?). If published, this will include your full peer review and any attached files.

Reviewer #1: No

Reviewer #2: **Yes: **Alimi Mustapha Faiz

---

## [Author Response · Author response to Decision Letter 0]

11 Jun 2024

Dear Editor and Reviewers:

We sincerely thank the editor and all reviewers for their valuable feedback that we have used to improve the quality of our manuscript. The reviewer comments are laid out below in italicized font and specific concems have been numbered. Our response is given in normal font and changes/additions to the manuscript are given in the revised manuscript with red text.

In response to the journal's requirements, we have revised and refined our manuscript, as detailed below: 

1. We have meticulously reviewed the journal's layout specifications and adjusted our manuscript accordingly. 

2. As per the guidelines, we have reformulated our declaration of interests and incorporated it into the cover letter.

3. We have thoroughly reviewed all references, ensuring that the sources are published, peer-reviewed articles from a variety of journals.

Editor Comments:

1.There needs to be greater emphasis on inclusion and exclusion criteria as well as a more thorough risk of bias assessment using PRISMA-endorsed instruments. 

2.The discussion also requires a greater overview of the literature in both a tabulated and written format. This should demonstrate and discuss other means of risk stratification in elderly patients undergoing surgery such as the CCI or MFI.

The author’s responses

1. We accept the editor's remarks and have incorporated the proposed changes into the document. To enhance the article's rigor, we have added two elements to the exclusion criteria after careful consideration. 1. Studies that used CGA solely as a risk assessment tool for postoperative unfavorable outcomes; 2. Studies that did not use a comprehensive multidomain evaluation and optimization plan, but only evaluated one CGA field, such as nutritional status. The newly generated inclusion and exclusion criteria are a more specific description of our search procedure and will not change our earlier search results. 

Furthermore, as the editor noted, PRISMA-approved instruments are essential for a thorough bias risk assessment. We used the Cochrane Collaboration's tool to assess the risk of bias in randomized trials and followed the meta-analysis guidelines for epidemiological observational research. Details can be found in the manuscript's quality evaluation section and the PRISMA checklist in the appendix.

2. Based on the suggestions, we explored other methods for preoperative risk stratification in elderly patients, such as the CCI and MFI. We summarized these methods in a table, including their names, brief descriptions, and areas of focus. Additionally, we elaborated on their connections and differences with CGA. Below are some discussion points from the revised manuscript.

Unlike CGA, these methods focus on clinical and physiological parameters that directly affect surgical risk, helping to decide whether patients should proceed with surgery and what level of care they need afterward. They provide faster evaluations but may miss subtle differences that a complete CGA can capture. CGA has a broader scope and is more widely used to assess the overall health status of patients. These aspects may not directly affect surgery but are crucial for patients' overall health and postoperative recovery.

Reviewer #1

1. Don't use acronyms for the first time. It's confusing.

2. Are the methods of CGA exactly the same? In the included original studies, if not, the authors may not be able to conduct a combined analysis.

3. The reasons for publication bias and high heterogeneity should be explained in the discussion.

4. Full text language needs to be modified.

The author’s responses

1. We apologize for the incorrect use of acronyms in the initial manuscript. Following the reviewer's suggestion, we have revised this section.

2. Regarding the reviewer's concerns about the heterogeneity of CGA methods, we addressed this during the literature review. CGA is a well-established multi-domain assessment that evaluates the physiological, social, psychological, and functional states of elderly patients. The approach to nursing optimization and the content addressed are consistent despite model variations. To achieve generalizable outcomes, we broadened the research criteria and decided to conduct a merger analysis. We carefully interpreted the results and acknowledged the limitations in the text, as this could introduce bias. However, recognizing and accepting the limitations of this review may encourage more scholars to conduct further research.

3. It is inevitable that bias will exist. We added the following potential sources of bias and heterogeneity to the manuscript's discussion section after summarizing them in response to reviewers' comments:

First, the potential for bias increased when the same researchers were involved in both the intervention and control groups, as the included studies were single-center and could not blind investigators and participants during the CGA intervention. Second, a time gap between pre- and post-operative evaluations could introduce extraneous variables. Some studies addressed this by shortening the observation period, leading researchers to select quickly examinable items. However, this may lead to publication bias. For example, studies on immediate postoperative delirium may be published earlier than those on long-term cognitive effects. Insufficient follow-up time can also result in false-negative outcomes. Third, this review is limited to English language research, and this limitation may also lead to bias.

4. Thanks for reviewers' significant reminding. We have asked native English speakers and tried our best to polish the manuscript. These changes will not influence the content and framework of the paper. And here we did not list the changes but marked in red in the revised paper.

Again, thank you for giving us the opportunity to strengthen our manuscript with your valuable comments and queries. We have worked hard to incorporate your feedback and hope that these revisions persuade you to accept our submission.

With best regards,

Sincerely Yours,

Corresponding author: Lin Chen

Corresponding address: No.10 Qingyun South Street, Jinjiang District, Chengdu City, Sichuan Province, China.

---

## [Editor Report · Decision Letter 1]

17 Jun 2024

The impact of comprehensive geriatric assessment on postoperative outcomes in elderly surgery: a systematic review and meta-analysis

PONE-D-24-12165R1

Dear Dr Chen,

We’re pleased to inform you that your manuscript has been judged scientifically suitable for publication and will be formally accepted for publication once it meets all outstanding technical requirements.

Kind regards,

Barry Kweh

Academic Editor

PLOS ONE

Additional Editor Comments (optional):

A well written article that has been amended in its methodology and discussion.
---

## [Editor Report · Acceptance letter]

21 Jun 2024

PONE-D-24-12165R1 

PLOS ONE

Dear Dr. Chen, 

I'm pleased to inform you that your manuscript has been deemed suitable for publication in PLOS ONE. Congratulations! Your manuscript is now being handed over to our production team.

Kind regards, 

on behalf of

Dr. Barry Kweh 

Academic Editor

PLOS ONE